# The Neural Mechanisms of Visual and Vestibular Interaction in Self-Motion Perception

**DOI:** 10.3390/biology14070740

**Published:** 2025-06-21

**Authors:** Jing Liu, Fu Zeng

**Affiliations:** 1Key Laboratory of Brain Functional Genomics (Ministry of Education), East China Normal University, Shanghai 200062, China; 52193200027@stu.ecnu.edu.cn; 2Key Laboratory of Brain Functional Genomics, Affiliated Mental Health Center (ECNU), School of Psychology and Cognitive Science, East China Normal University, Shanghai 200062, China

**Keywords:** multisensory integration, self-motion perception, visual–vestibular interaction, Bayesian inference, neural computation

## Abstract

The brain maintains a stable sense of self-motion by flexibly combining visual information (optic flow) with vestibular signals from the inner ear’s motion sensors. This integration allows us to move, walk, and balance effectively in everyday life. When visual and vestibular cues disagree—such as during virtual reality, turbulent motion, or certain neurological conditions—people can experience motion sickness, disorientation, or perceptual disturbances, which are also observed in disorders like autism and Parkinson’s disease. Research indicates that the brain uses a near-optimal, reliability-weighted averaging strategy: it dynamically assesses the trustworthiness of each cue and adjusts their influence accordingly. Understanding these mechanisms not only clarifies how we navigate complex environments but also informs potential interventions for conditions involving impaired sensory integration. This review synthesizes behavioral, neurophysiological, computational, and imaging findings to highlight how dynamic visual–vestibular integration underpins accurate self-motion perception and may guide therapeutic approaches for sensory integration disorders.

## 1. Introduction

Visual–vestibular integration plays a fundamental role in enabling the brain to construct accurate and stable representations of self-motion and spatial orientation [1,2,3,4]. By combining information from the visual system—particularly optic flow patterns—with signals from the vestibular system that encode head motion and acceleration, the brain achieves a coherent perception of body movement in space. This multisensory process is essential in maintaining perceptual stability, guiding navigation, and supporting posture and balance during everyday behaviors.

When visual and vestibular signals are not properly integrated, the resulting sensory conflict can lead to perceptual instability and disorientation. Such integration failures are associated with common conditions including motion sickness, spatial disorientation, and vestibular disorders [5,6,7]. Moreover, abnormal multisensory processing has been reported in various neurological and psychiatric conditions—such as autism spectrum disorder [8,9,10], schizophrenia [11], and Parkinson’s disease [12]—suggesting that atypical sensory integration may be related to these disorders.

Early research on multisensory integration between vestibular and visual signals primarily focused on behavioral phenomena. With increasing interest in the underlying processes, recent studies have progressively shifted their attention toward the neural mechanisms supporting this integration. In this review, we use visual–vestibular interactions in self-motion perception as a model system to explore the behavioral principles and neural substrates of visual–vestibular integration.

## 2. Self-Motion Perception Involving Visual–Vestibular Integration

Self-motion perception refers to the ability to detect one’s movement and changes in head position relative to the external world. In everyday life, accurate self-motion perception is vital for survival, as it allows individuals to make appropriate judgments and plan actions based on their dynamic relationship with the environment. This process relies on the integration of multiple sensory signals, among which visual and vestibular inputs are the two primary sources of information [13].

Visual signals contribute through optic flow—the systematic pattern of motion across the retina, generated as we move through our environment. Optic flow provides critical information about the relative movement between the observer and the surrounding world [14]. Numerous psychophysical studies have investigated how the brain estimates the heading direction based on the analysis of optic flow [15,16,17,18].

Vestibular signals, on the other hand, are derived from sensors located in the inner ear: the semicircular canals, which detect angular acceleration, and the otolith organs, which detect linear acceleration, including the effects of gravity [19,20,21,22,23]. These sensors continuously monitor the position and movement of the head, thereby supplying direct information about self-motion [24,25,26,27]. Beyond its contribution to basic self-motion detection, the vestibular system plays a central role in ensuring perceptual stability and guiding accurate motor responses. It does so not only by relaying sensory information about head motion but also by engaging predictive mechanisms that distinguish between self-generated and externally applied motion, thereby stabilizing visual perception during active behaviors [28]. Furthermore, vestibular signals are essential for gaze stabilization via the vestibulo-ocular reflex and in maintaining posture and balance through reflexive and voluntary pathways [29]. Importantly, vestibular inputs also reach widespread cortical and hippocampal regions, where they contribute to higher-level cognitive functions such as spatial orientation, memory, and navigation [30]. These integrative roles position the vestibular system as a key sensory modality not only for motion detection but in coordinating action and supporting spatial cognition.

However, in naturalistic environments, optic flow information is often ambiguous or disrupted. Rotations of the head or eyes [31,32], as well as the relative motion of other objects in the visual field [33,34], can distort the structure of the optic flow, complicating the brain’s interpretation of self-motion. The vestibular system also has inherent limitations—it cannot detect constant-velocity linear motion and struggles to distinguish between head tilt caused by gravity and translation relative to the ground [35]. However, studies have shown that the nervous system addresses this equivalence problem by integrating otolith and semicircular canal signals at multiple levels of the central nervous system [36,37,38,39]. In addition, multisensory integration provides a robust solution: biological systems tend to integrate visual and vestibular information to achieve more reliable self-motion perception. This visual–vestibular integration not only compensates for the weaknesses of each individual system but also significantly enhances the accuracy and stability of motion perception [3,4,40,41,42,43,44,45].

## 3. Bayesian Optimal Computational Model of Visual–Vestibular Integration

To achieve accurate and stable self-motion perception, the brain needs to integrate information from different sensory channels. Computational theories provide a mathematical framework for this problem (maximum likelihood estimation) [1,46,47,48,49], with the fundamental concept being that sensory information inherently carries uncertainty [50], and the encoding of this information by the sensory systems (such as the firing rates of visual/vestibular neurons) is also stochastic [2,51,52,53,54,55].

To estimate the self-motion heading direction S, visual cue $C_{vis}$ and vestibular cue $C_{vest}$ each provide representations of S that can be described by normal distributions $p\left(C_{vis}|S\right)$ and $p\left(C_{vest}|S\right)$, respectively. The observer’s belief about the stimulus S can be expressed through the likelihood function $L\left(S\right)\equiv p\left(C|S\right)$. Assuming that the noise of the two sensory modalities is independent, the likelihood function for the combined visual–vestibular condition can be decomposed into the product of individual cue likelihoods:(1)$$L_{com}\left(S\right)\equiv p\left(C_{vis},C_{vest}|S\right)=p\left(C_{vis}|S\right)p\left(C_{vest}|S\right)=L_{vis}\left(S\right)L_{vest}\left(S\right)$$

The Bayesian perceptual model introduces a prior $p\left(S\right)$, and the posterior probability of the current heading S, based on synchronous visual and vestibular cues, is given by(2)$$p\left(S|C_{vis},C_{vest}\right)\propto \;p\left(C_{vis},C_{vest}|S\right)\cdot p\left(S\right)\equiv p\left(C_{vis}|S\right)p\left(C_{vest}|S\right)\cdot p\left(S\right)$$

By maximizing the posterior distribution $p\left(S|C_{vis},C_{vest}\right)$, we can obtain the best estimate of heading S^. When the prior $p\left(S\right)$ is flat, the posterior is determined by the likelihood function $p\left(C_{vis},C_{vest}|S\right)$, and the mean μ and variance $\sigma^{2}$ of the distribution of S^ satisfy(3)$$\mu_{com}=w_{vis}\mu_{vis}+w_{vest}\mu_{vest}=\frac{r_{vis}\mu_{vis}+r_{vest}\mu_{vest}}{r_{vis}+r_{vest}}$$

Here, w represents the weights of different sensory modalities during integration, and r represents the reliability of each cue (defined as the inverse of the variance $1\;/\sigma^{2}$). The reliability of the combined estimate is the sum of the reliabilities of each cue:(4)$$\frac{1}{\sigma_{com}^{2}}=\frac{1}{\sigma_{vis}^{2}}+\frac{1}{\sigma_{vest}^{2}}$$

The Bayesian optimal integration model linearly weights the information according to its reliability, with the combined cue’s reliability being superior to that of any single cue (thus reducing uncertainty), and it tends to favor cues with higher reliability [1,2,56,57]. The Bayesian model is capable of handling the uncertainty in sensory information and better explains behavioral performance through top-down expectations or stimulus biases. Whether these biases are acquired through task learning or established by life experience, the Bayesian model effectively integrates this knowledge into perceptual decision-making.

In this framework, not only does multisensory integration depend on the reliability of sensory inputs, but it is also closely linked to the individual’s prior beliefs [58]. For instance, in situations where visual and vestibular cues are in conflict, the Bayesian model can dynamically adjust the weights of visual and vestibular information based on their relative reliability, thereby achieving an optimal perceptual outcome [4]. Recent studies reveal that the brain’s internal models of gravity significantly influence multisensory integration mechanisms, with particularly strong effects on vertical motion perception—findings that align with Bayesian models of perceptual inference [59,60].

This Bayesian perceptual model has been applied in several behavioral experiments, confirming how the brain optimizes the integration of information to enhance perceptual accuracy and stability [3,4,51,61,62]. Particularly when facing multiple sensory inputs, the predictions of the Bayesian model align with experimental data, indicating that the brain indeed follows this theoretical framework during perceptual processes.

## 4. Behavioral Research on Visual–Vestibular Integration

A large body of human psychophysical research has demonstrated that, when multiple sensory cues are available, observers typically exhibit behavior consistent with Bayesian optimal integration [51,54,63]. These studies show that the brain tends to weight each cue according to its reliability, resulting in perceptual estimates that are more precise than those based on individual cues alone.

In non-human primates, Gu and colleagues conducted a seminal study investigating visual–vestibular integration using a heading discrimination task. In this two-alternative forced choice (2AFC) paradigm, macaques were trained to report whether their perceived heading direction was to the left or right of straight ahead by making a corresponding saccade. Across trials, the monkeys were presented with visual cues (optic flow), vestibular cues (inertial motion), or both cues simultaneously.

The results showed that heading perception was significantly more accurate under the combined condition than under either unimodal condition. Psychometric functions revealed lower discrimination thresholds in the combined condition, consistent with the predictions of a Bayesian optimal integration model (Figure 1A). These findings provide strong behavioral evidence that the primate brain integrates visual and vestibular inputs in a statistically optimal fashion to improve perceptual precision.

To further explore the dynamic effects of cue reliability on heading perception, a study built upon Gu et al.’s experiment by introducing bimodal conditions with visual–vestibular heading conflicts and varying the visual coherence gradient (to alter the weight of visual cues) [64]. The results showed that, when visual coherence was higher, the subjects’ heading perception tended to favor the visual direction. As visual coherence decreased, the visual weight was reduced while the vestibular weight increased, causing the heading perception to shift toward the vestibular direction. These findings are consistent with the predictions of the Bayesian integration model (Figure 1B).

Although numerous studies have shown that the integration of visual and vestibular information enhances the precision of motion perception [4,40], this integration is most effective when the optic flow and vestibular signals are aligned or nearly aligned in direction. Therefore, the brain must infer the causal relationships behind these cues to handle multisensory information in cases of conflict [65,66,67,68]. Some studies have analyzed how explicit and implicit causal inference strategies are employed to integrate information when there are differences between visual and vestibular cues. In explicit causal inference tasks, subjects are required to explicitly judge whether visual and vestibular cues originate from the same source, while, in implicit causal inference tasks, subjects only need to judge the vestibular direction, implicitly assessing the causal relationship between visual and vestibular cues to form the final directional perception. Such research shows that the brain does not simply merge the two cues but adjusts the perceptual strategy based on their offsets. When the offsets between the two cues are small, the brain tends to integrate them; when the offsets are large, the brain tends to separate the cues that represent different directions [68].

Moreover, when visual and vestibular cues are inconsistent, the brain dynamically adjusts the perceptual discrepancies between the two in order to maintain perceptual consistency and accuracy. This process is known as multisensory recalibration. In self-motion perception, visual–vestibular recalibration has two mechanisms: unsupervised and supervised recalibration [69,70,71]. In the absence of external feedback, unsupervised recalibration causes the perception of visual and vestibular cues to adjust in opposite directions to reduce the conflict between them, achieving “internal consistency”. Under these conditions, the psychometric curve shifts in the opposite direction (Figure 1C). When there is a difference between visual and vestibular cues, providing feedback on the accuracy of one of the cues (such as vestibular or visual) will trigger supervised recalibration. This recalibration aims for “external accuracy”, where the comparison between the cues and environmental feedback leads to “yoked adjustment”, causing the psychometric curve to shift in the same direction (Figure 1D).

The integration of visual and vestibular information plays a crucial role in self-motion perception, and this process aligns with the predictions of the Bayesian optimal integration model. Studies show that the brain dynamically adjusts its integration strategy based on the reliability of each sensory cue to improve the perceptual accuracy. However, in cases of conflicting information, the brain does not merely perform weighted averaging but instead dynamically adjusts the perceptual strategy through causal inference to ensure perceptual consistency and accuracy. Additionally, the visual–vestibular recalibration mechanism further reveals how the brain adapts based on external feedback or internal cue differences, ensuring precise motion perception in a complex multisensory environment. These studies provide profound behavioral evidence for the neural mechanisms of perceptual integration and offer important insights into the brain’s adaptive adjustments in complex perceptual situations.

## 5. Visual–Vestibular Integration Neural Network

### 5.1. Brain Regions Involved in Visual–Vestibular Integration

To explore the neural mechanisms underlying visual–vestibular integration, numerous studies have focused on brain regions that exhibit selectivity for the heading direction. Figure 2 illustrates the pathways for the transmission of vestibular and visual information, along with the cortical network potentially involved in visual–vestibular heading perception [72,73,74,75,76,77,78].

In self-motion perception, vestibular signals first project to the vestibular nuclei (VN) and then travel via the vestibulo-thalamic pathway directly to the parieto-insular vestibular cortex (PIVC) [79,80,81]. From here, vestibular information is relayed to other multisensory areas, such as the dorsal medial superior temporal area (MSTd) [82,83,84,85,86], the ventral intraparietal area (VIP) [87,88,89,90,91,92,93], the visual posterior sylvian area (VPS) [94,95], the smooth eye movement area of the frontal eye field (FEF_sem_) [96], the superior temporal polysensory area (STP) [97], and area 7a [98].

Visual motion signals, such as optic flow, are first relayed through the lateral geniculate nucleus (LGN) to the primary visual cortex (V1) and subsequently processed in the middle temporal area (MT) along the dorsal visual pathway [79]. The receptive fields of neurons in V1 [99,100] and MT [101,102] are relatively small and primarily handle local motion information. Visual signals are then projected to MSTd, where neurons have larger receptive fields and can extract global motion information [82,83,103], before being passed to other multisensory brain regions.

Among these areas, the MSTd serves as a critical node where visual and vestibular information converges. In macaques, neurons in the MSTd respond to both visual and vestibular stimuli, with a pronounced preference for specific directions [82,85]. By comparing MSTd neurons’ tuning to visual and vestibular directions, researchers have classified neurons into two categories: “congruent” neurons, which have aligned preferences for both visual and vestibular stimuli, and “opposite” neurons, which have opposing directional preferences [85]. Congruent neurons’ activity is closely related to behavior involving direction discrimination [25], supporting the idea that vestibular information contributes to heading perception in the MSTd. Opposite neurons, while their specific function remains unclear, may be involved in distinguishing self-motion from object motion [104]. Studies have shown that microstimulation in the MSTd can significantly alter self-motion perception based on optic flow [105,106], although it has minimal effects on vestibular motion direction judgments in darkness [105]. Additionally, the reversible inactivation of the MSTd affects visual direction judgments but has little impact on vestibular perception, highlighting the MSTd’s crucial role in processing optic flow for self-motion perception. In contrast, vestibular conditions rely more on other cortical areas, such as the PIVC or VPS [90,94].

In humans, the MT+ complex includes the hMT and hMST, delineated using retinotopic mapping and receptive field asymmetries [107,108]. Galvanic vestibular stimulation (GVS) studies show that vestibular responses are confined to the hMSTa, the anterior portion of the hMST [77], consistent with the macaque MSTd. Congruent and opposite visual–vestibular roll stimuli have been shown to elicit distinguishable multivoxel activation patterns in the hMSTa, as revealed by multivoxel pattern analysis [109]. This suggests the existence of functionally distinct neuronal populations within the hMSTa, potentially analogous to the congruent and opposite cells found in the macaque MSTd. Although direct evidence for vestibular heading encoding in the hMSTa remains lacking, these findings imply close homology to the MSTd.

The PIVC is considered the core area for the processing of vestibular information. Most neurons in the macaque PIVC do not respond to visual stimuli [110], which suggests that the PIVC is not the main site for visual–vestibular integration. However, a distinct cortical area posterior to the PIVC, known as the VPS, responds to both rotation and translation stimuli in the dark [94]. Unlike the PIVC, the VPS also shows significant responses to optic flow [94]. The VPS is closely connected to the PIVC, receiving vestibular signals and possibly optic flow signals from the MSTd [81]. Many neurons in the macaque VPS are tuned to both visual and vestibular motion directions. As with the MSTd, the VPS also contains congruent and opposite neurons. However, the dominant population in the VPS is opposite neurons, suggesting that the VPS plays a key role in visual–vestibular interactions.

In humans, the PIVC has been identified as receiving similar subcortical vestibular inputs [111,112,113,114]. The posterior parietal operculum, particularly area OP2, has been proposed as a key node for visual–vestibular integration in humans [115]. OP2 is homologous to the macaque PIVC and is consistently activated by vestibular stimulation [114,116,117]. It receives converging vestibular, somatosensory, and visual inputs [81], and plays a crucial role in self-motion perception [118]. While the anterior portions of OP2 primarily process vestibular signals, adjacent retroinsular regions also respond to visual motion, suggesting a spatial gradient of multisensory integration within the broader OP2+ complex [119]. Thus, OP2 and its neighboring regions may serve as an important interface for the processing and integration of multisensory cues related to self-motion.

Accumulating evidence suggests that the posterior insular cortex (PIC) may serve as the human homolog of the macaque VPS [120]. Functional MRI studies have shown that the PIC responds to both vestibular and visual stimuli. When the two modalities are presented together, responses to congruent and opposite motion directions resemble the response patterns observed in the VPS [119,121,122]. Moreover, the structural connectivity profile of the PIC supports its proposed role in visual–vestibular processing: it is connected with regions such as the superior temporal sulcus, intraparietal cortex, and supramarginal gyrus [123,124]. These findings suggest that the PIC may play a central role in human visual–vestibular integration, potentially mirroring the function of the VPS in macaques.

Recent human studies have shown that several medial cortical regions are involved in visual–vestibular processing underlying self-motion perception [125,126], with the cingulate sulcus visual area (CSv) being a particularly notable example. The CSv responds robustly to optic flow stimuli but not to random motion or static visual input [127]. It also exhibits strong responses to artificial vestibular stimulation, indicating that it is a multisensory region engaged in both visual and vestibular processing [77]. However, current evidence suggests that the CSv does not integrate these sensory signals. Instead, it may relay them to motor-related regions, as it shows strong connectivity with areas such as the supplementary motor area [128].

The VIP, located at the bottom of the intraparietal sulcus, is another multisensory integration area. The majority of the neurons in the macaque VIP show significant responses to both optic flow and vestibular stimuli [87,88,92]. The response characteristics of the VIP to optic flow are similar to those of the MSTd, with strong directional selectivity and a neural connection between the two regions [129], indicating that the MSTd may be the primary source of the VIP’s visual input [90]. An analysis of the direction preferences of VIP neurons revealed the presence of both congruent and opposite neurons [89,90], similarly to the MSTd and VPS. However, unlike the MSTd, the activity of VIP neurons shows a stronger correlation with directional judgment [91]. Despite this, causal manipulation experiments show that the inactivation of the VIP does not significantly affect directional perception based on visual or vestibular cues, and microstimulation in the VIP has no significant effect on optic flow-based direction perception [118]. Zaidel, DeAngelis [130] further demonstrated that, in the VIP, choice-related (top-down) signals are dominant, while, in the MSTd, sensory-driven (bottom-up) signals prevail; see also Chen, Zeng [131].

The human VIP (hVIP), located in the fundus of the intraparietal sulcus, responds to visual, auditory, and somatosensory inputs [112,132]. It is more sensitive to self-motion-compatible optic flow than to spatially fragmented motion [126]. Although vestibular responses are weak or undetectable in the hVIP [77], decoding analyses have shown that hVIP activity distinguishes between congruent and opposite vestibular–visual stimuli [109]. This suggests the presence of latent vestibular sensitivity not easily captured by standard fMRI.

In addition to the aforementioned areas, multisensory brain regions such as the FEF_sem_, 7a, and the STP are also involved in visual–vestibular integration. The FEF_sem_, associated with the control of smooth eye movements, shows directionally selective responses to both visual and vestibular stimuli [96]. Similarly to the MSTd and VIP, FEF_sem_ neurons exhibit similar directional preferences for visual and vestibular modalities, with both congruent and opposite neurons present. However, the FEF_sem_ is primarily involved in eye movement control rather than perception, suggesting that, during self-motion perception, visual–vestibular interactions may serve more to coordinate eye movements and maintain visual stability [96]. Area 7a is highly sensitive to both visual and vestibular cues, with its response to visual–vestibular joint stimuli typically dominated by vestibular input, even suppressing visual input in some cases. This suggests that 7a may be more involved in resolving cue conflicts rather than integrating these cues for self-motion estimation [98]. The STP, which shows directionally selective responses to both visual and vestibular information, is more visually dominant [97]. Its vestibular and visual tuning strength is weaker than in the MSTd, VIP, and VPS, indicating that the STP plays a lesser role in self-motion perception [97].

Together, these multisensory brain regions orchestrate the integration of vestibular and visual signals, enabling accurate and adaptive self-motion perception. However, how neurons integrate sensory inputs from different modalities to generate self-motion perception remains an open question. The following discussion will focus on the computational rules governing the integration of visual–vestibular information by neurons—specifically, how interactions between neurons and computational strategies enable efficient perception integration and adaptive adjustments at the brain level.

### 5.2. Computational Implementation and Neural Mechanisms of Visual–Vestibular Integration

In self-motion perception, the integration of visual and vestibular signals follows a Bayesian optimal computational model. However, a question arises: what computational rules do neurons follow to implement this process?

Early work explored the neural encoding mechanisms behind visual–vestibular integration through a series of experiments. For instance, Gu, Angelaki [3] recorded the activity of MSTd neurons while macaques performed a heading discrimination task involving visual and vestibular cues, finding that macaques could estimate the heading direction based on optimal cue combination. An analysis of neuron activity through ROC curves revealed that the “congruent” neurons in the MSTd exhibited a visual–vestibular discrimination threshold that aligned with optimal cue combination predictions. Moreover, in combined conditions, the sensitivity of these neurons to changes in the translation direction significantly increased [3]. Similar results were observed in the VIP area, where neuronal activity also followed the Bayesian optimal integration model [91].

However, early studies focused on relatively small angular ranges (left and right deviations from the forward direction) during heading discrimination tasks in macaques. Moreover, factors like choice could affect neuronal activity and integration rules, preventing the clear depiction of a neural combination rule. To address this limitation, Morgan, DeAngelis [86] presented macaques with full-range visual, vestibular, and visual–vestibular combined stimuli covering a 360° horizontal plane and recorded the MSTd neuron activity. Their results showed that a linear weighted model could well describe the relationship between neural responses under bimodal conditions and responses to unimodal visual and vestibular stimuli:(5)$$r_{comb}=w_{vis}r_{vis}+w_{vest}r_{vest}$$
where $w_{vis}$ and $w_{vest}$ are weights less than 1, indicating that visual and vestibular inputs are typically subadditive rather than additive. As the visual coherence decreased, the weights for visual and vestibular inputs shifted systemically: the visual weight increased, while the vestibular weight decreased. Fetsch and colleagues [64] further demonstrated that, even with random changes in reliability across trials, neuronal adjustments to weights still occurred, suggesting that neurons may dynamically adjust their combination rules based on cue reliability. Ohshiro and colleagues [133] proposed a divisive normalization model at the level of population neurons, where the output of each neuron is divided by the total activity of a normalization pool. This pool could consist of all neurons within a functional brain area, neurons within a certain distance from the target neuron, or neurons that share similar stimulus feature preferences. This normalization mechanism is conceptually similar to lateral inhibition, except that lateral inhibition typically involves subtraction, while normalization adjusts neuronal responses via division. The neuronal response *R* after normalization can be expressed as(6)$$R=\frac{E^{n}}{\alpha^{n}+\frac{1}{N}\sum_{i=1}^{N}E_{i}^{n}}$$
where $E^{n}$ represents the sensory input to a multisensory neuron, increasing sublinearly with the stimulus intensity (with exponent n), and $E_{i}$ represents the total activity of neurons in the normalization pool. $\alpha^{n}$ is a constant that determines how neurons respond to increases in stimulus intensity.

The divisive normalization mechanism provides a theoretical framework for an understanding of the interaction of different sensory signals in multisensory brain areas. Through this mechanism, the changes in neuronal responses can effectively explain the neural combination rules observed in MSTd experiments [64,86]. In this model, the strength of the normalization pool depends on the discharge rates of individual sensory neurons. Specifically, under visual conditions, the total sum of the normalization signal varies significantly with changes in motion coherence. However, under combined conditions, vestibular signals contribute in a way that is independent of coherence, which weakens the impact of coherence on the normalization pool. This phenomenon helps to explain the rapid shifts in neuronal response weights observed across trials. Through the division-based normalization mechanism, neuronal responses are restricted, avoiding saturation effects and also explaining the subadditivity of neural weights (i.e., w < 1).

Recent studies have further examined how visual and vestibular cues interact in multisensory cortical areas such as the MSTd and VPS. Results show that, when vestibular and visual cues are congruent, MSTd neuronal responses are enhanced across modalities. In contrast, when these cues are in conflict, neuronal responses are suppressed, and this suppression increases as the disparity between the preferred directions of visual and vestibular inputs grows [95]. These findings are consistent with previous observations suggesting that inputs from non-preferred modalities can suppress responses from preferred ones [134], and they align with the predictions of the divisive normalization model (Figure 3A). In the VPS, the introduction of different directional visual stimuli suppresses vestibular responses. As the visual stimulus direction deviates from the preferred vestibular direction, the cross-modal suppression effect becomes stronger [95] (top row, Figure 3B).

A linear weighting model was used to fit neuronal responses in the MSTd and VPS under bimodal stimulation. The model revealed that visual inputs had higher weights in the MSTd, while vestibular inputs dominated in the VPS. This indicates that the MSTd relies more on visual information for integration, whereas the VPS primarily depends on vestibular cues.

Despite these distinct profiles, the cross-modal interaction patterns in both areas can largely be explained by the normalization model. When presented with combined stimuli, MSTd neurons exhibit a subadditive response pattern, reflecting cross-modal enhancement. In contrast, VPS neurons demonstrate a strong “winner-takes-all” mechanism: when visual signals are added to vestibular stimulation, population responses are largely dominated by vestibular inputs, while visual signals exert cross-modal suppression—even in neurons that are highly responsive to visual cues alone. These findings suggest that neuronal responses in these areas are shaped by broader network-level interactions and that divisive normalization may serve as a general computational principle governing visual–vestibular integration (Figure 4).

In summary, visual–vestibular integration involves a sophisticated network of brain regions, each playing a specific role in processing sensory information and adjusting weights based on reliability. Through mechanisms like normalization and dynamic sensory weighting, the brain achieves accurate self-motion perception even in complex and noisy environments. These neural networks and computational models provide important insights into how the brain efficiently combines multiple sensory inputs for adaptive behavior.

## 6. Conclusions

In this review, we have comprehensively outlined the crucial role of multisensory integration in self-motion perception, with a particular focus on the interaction between visual and vestibular signals. By exploring how the brain integrates these two sensory modalities, we have discussed the theoretical framework of the Bayesian optimal integration model that underlies the neural computational mechanisms behind this process. We have further explored the cortical networks involved in processing optic flow and vestibular cues—such as the MSTd and VPS—highlighting the computational rules that govern visual–vestibular integration.

However, the neural mechanisms underlying visual–vestibular integration in other multisensory areas, such as the VIP and FEF_sem_, remain poorly understood. Future studies are needed to investigate the interactions between neurons in these areas, with particular attention to the neural basis of causal inference—a general computational principle that allows the brain to attribute observed outcomes to potential causes. In addition, further work is required to examine how conflicts between visual and vestibular information affect multisensory processing, which may deepen our understanding of how the brain integrates sensory signals in an efficient and adaptive manner.

In conclusion, research on visual–vestibular integration provides valuable insights into the fundamental principles of multisensory perception and has important implications for neurocognitive models of perception, learning, and behavior. By continuing to unveil the computational strategies and neural networks involved, future research will further advance our understanding of the brain’s adaptive capabilities and provide crucial references for the development of interventions and technologies aimed at addressing sensory integration disorders.

## Figures and Tables

**Figure 1 biology-14-00740-f001:**
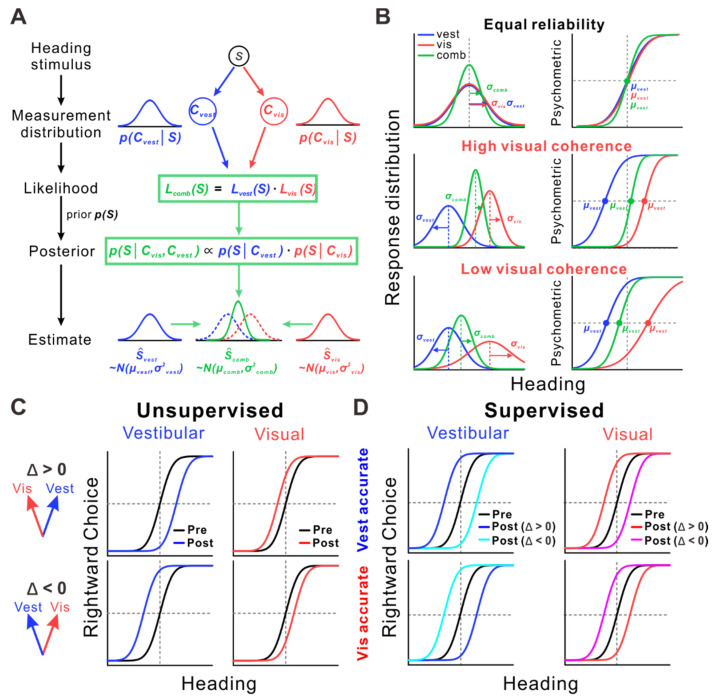
Bayesian optimal integration model (**A**,**B**) and recalibration processes in multisensory perception (**C**,**D**). (**A**) A heading stimulus S generates both visual and vestibular cues, which are modeled as independent normal distributions, $p\left(C_{vis}|S\right)$ and $p\left(C_{vest}|S\right)$, respectively. According to the maximum likelihood estimation principle, the combined likelihood function can be expressed as the product of the individual cue likelihoods. In the Bayesian framework, a prior distribution $p\left(S\right)$ is introduced; assuming a flat prior, the posterior is determined solely by the combined likelihood. The optimal heading estimate S^ is obtained by maximizing the posterior distribution. (**B**) Posterior probability distributions (left) and example psychometric functions from heading discrimination task (right). Top row: When visual and vestibular cues are equally reliable, optimal integration results in enhanced perceptual precision. Middle row: When the two cues are in conflict with high visual reliability (high coherence), the combined estimate is biased toward the visual cue. Bottom row: When the two cues are in conflict with low visual reliability (low coherence), the combined estimate is biased toward the vestibular cue. (**C**) Unsupervised recalibration. The black curve indicates psychometric performance before recalibration. After recalibration, the vestibular (blue) and visual (red) heading estimates are shifted. When Δ > 0, the vestibular and visual estimates shift rightward and leftward, respectively; when Δ < 0, the vestibular and visual estimates shift leftward and rightward, respectively. Psychometric functions exhibit opposite shifts for the two modalities. (**D**) Supervised recalibration. The top and bottom panels correspond to conditions where the “vestibular accurate” and “visual accurate” cues are designated, respectively. Here, “accurate” indicates consistency between the cue and external feedback. In both conditions, the psychometric curves shift in the same direction toward the modality identified as accurate.

**Figure 2 biology-14-00740-f002:**
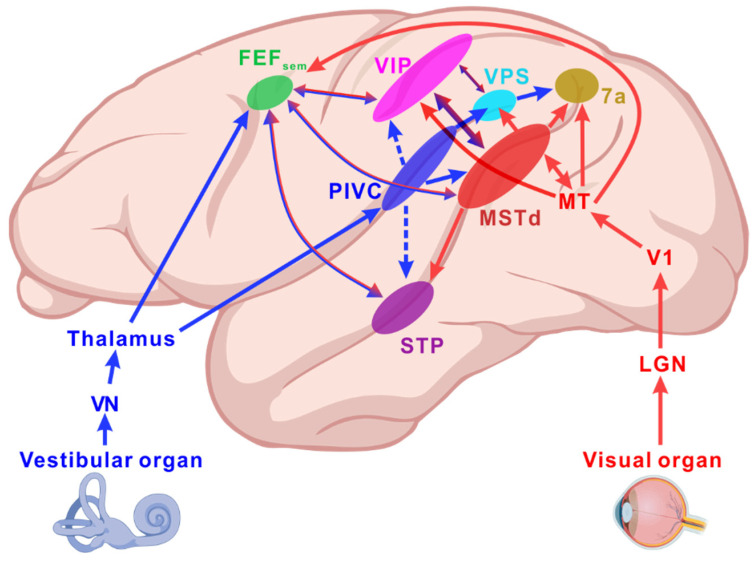
Vestibular and visual pathways in self-motion perception. The blue arrow represents the vestibular information pathway: peripheral vestibular signals are first transmitted to the vestibular nucleus (VN) and then relayed through the thalamus, primarily projected to the parieto-insular vestibular cortex (PIVC), and may further extend to the frontal cortex. The red arrow represents the visual information pathway: visual information from the retina is transmitted through the lateral geniculate nucleus (LGN) to the primary visual cortex (V1) and subsequently projected through the middle temporal area (MT) to other visual cortical regions involved in motion direction perception. MSTd, dorsal portion of medial superior temporal area; PIVC, parieto-insular vestibular cortex; VPS, visual posterior sylvian area; VIP, ventral intraparietal area; FEF_sem_, smooth eye movement region of the frontal eye field; STP, superior temporal polysensory area.

**Figure 3 biology-14-00740-f003:**
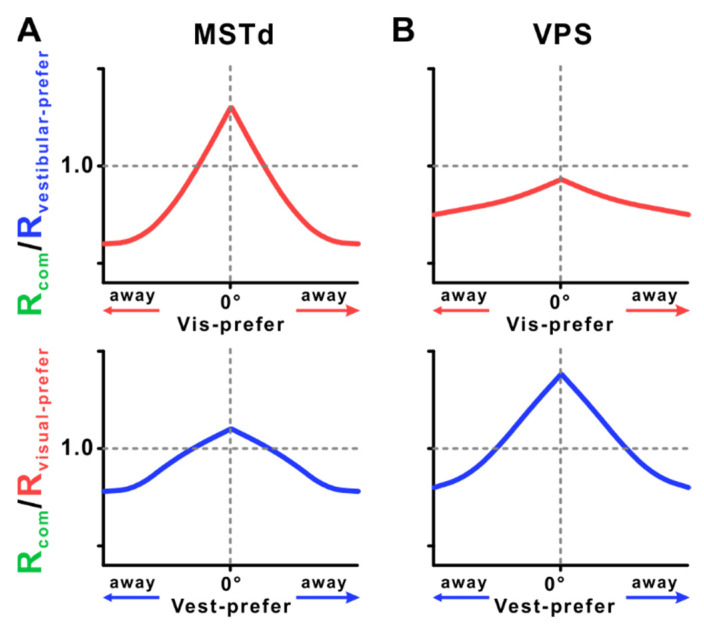
Modulation of neuronal responses by added sensory inputs across MSTd (**A**) and VPS (**B**). (**Top**): The red line represents the ratio of the combined response (*R_comb_*) to the vestibular-only response at the vestibular preferred direction (*R_vestibular-prefer_*) as a function of the added visual stimulus direction (0° indicates the neuron’s visual preferred direction). A ratio greater than 1 indicates that adding the visual stimulus enhances the vestibular response, while a ratio less than 1 indicates suppression. (**Bottom**): The blue line represents the ratio of the combined response (*R_comb_*) to the visual-only response at the visual preferred direction (*R_visual-prefer_*) as a function of the added vestibular stimulus direction (0° indicates the neuron’s vestibular preferred direction). A ratio greater than 1 indicates facilitation by the vestibular stimulus, whereas a ratio less than 1 indicates suppression.

**Figure 4 biology-14-00740-f004:**
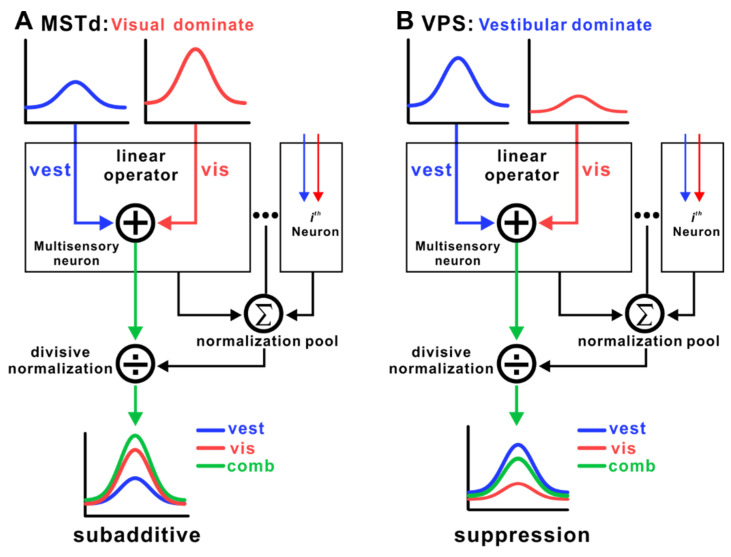
Computational mechanisms of vestibular and visual signal integration across MSTd and VPS. (**A**) In the MSTd, multisensory neurons linearly integrate vestibular (blue) and visual (red) inputs, with visual signals typically dominating. The integrated signals are subsequently subject to divisive normalization via a normalization pool, resulting in a subadditive combined response (green), where the response to multisensory inputs is weaker than the sum of unimodal responses. (**B**) In the VPS, vestibular inputs are dominant, and divisive normalization causes the suppression of the combined response relative to the vestibular response alone.

## Data Availability

No new data were created or analyzed in this study.

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
