# Peer review of "The Neural Mechanisms of Visual and Vestibular Interaction in Self-Motion Perception"

_biology, 2025, doi:10.3390/biology14070740_

Round 1
Reviewer 1 Report
Comments and Suggestions for Authors
This is a very nice, timely review of a complex issue, the neural basis of visual-vestibular integration for self-motion.
In general, the main topics are all well discussed and pertinent literature is cited.
However, there might be some specific integrations that the Authors may want to consider.
I list them following the line numbering of the article.
Line 74. “The vestibular system…. struggles to distinguish between head tilt caused by gravity and translation relative to the ground (Angelaki, 2004)”
In fact, work from Angelaki’s group has unraveled some of the basic mechanisms to distinguish head tilt due to gravity and linear translation, and the Authors may want to cite some pertinent work, e.g. Angelaki, D.E., Shaikh, A.G., Green, A.M. and Dickman, J.D., 2004. Neurons compute internal models of the physical laws of motion. Nature, 430(6999), pp.560-564. Laurens, J., Meng, H. and Angelaki, D.E., 2013. Neural representation of orientation relative to gravity in the macaque cerebellum. Neuron, 80(6), pp.1508-1518.
Therefore, it should be clarified that the equivalence problem of gravitational and linear acceleration is addressed by the nervous system not only by integrating visual and vestibular cues, but also by integrating otolith signals and semicircular canals at different stages of the central nervous system (behaviorally, this was already suggested in humans by Merfeld, D.M., Zupan, L. and Peterka, R.J., 1999. Humans use internal models to estimate gravity and linear acceleration. Nature, 398(6728), pp.615-618, as well as by Angelaki, D.E., McHenry, M.Q., Dickman, J.D., Newlands, S.D. and Hess, B.J., 1999. Computation of inertial motion: neural strategies to resolve ambiguous otolith information. Journal of Neuroscience, 19(1), pp.316-327.)
Line 82. This section develops very thoroughly the issue of Bayesian integration. The Authors may want to consider adding this nice review on the specific topic: Angelaki, D.E., Klier, E.M. and Snyder, L.H., 2009. A vestibular sensation: probabilistic approaches to spatial perception. Neuron, 64(4), pp.448-461.
In addition, they may want to consider the issue of visual and vestibular processing of vertical motion by taking into account the effects of gravity: Delle Monache, S., La Scaleia, B., Finazzi Agrò, A., Lacquaniti, F. and Zago, M., 2025. Psychophysical evidence for an internal model of gravity in the visual and vestibular estimates of vertical motion duration. Scientific Reports, 15(1), p.10394.
Line 115. “multisensory integration not only depends on the reliability of sensory inputs but is also closely linked to…” It would be better: “not only does multisensory integration depend on…, but it is also..”
Line 227 ref. 47 should be cited since Guldin and Grusser were the first to suggest the term PIVC, at least to the best of my knowledge
The following lines of the section Brain Regions Involved in Visual-Vestibular Integration give an excellent coverage of the main literature based on work in non-human primates.
However, I suggest that the Authors consider expanding the coverage of literature in humans, since the results may not necessarily align with those in non-human primates.
First, they may want to consider a review comparing monkey and human data. The latter are mainly (but not exclusively, there is also work base on patients) based on neuroimaging for the anatomo-functional substrates in humans. Smith, A.T., Greenlee, M.W., DeAngelis, G.C. and Angelaki, D.E., 2017. Distributed visual–vestibular processing in the cerebral cortex of man and macaque. Multisensory Research, 30(2), pp.91-120.
In particular, the question of the human homologue of monkey PIVC has been much debated.
VPS of monkey could be PIC (Sunaert S, Van Hecke P, Marchal G, Orban GA. Motion-responsive regions of the human brain. Exp Brain Res 127: 355–370, 1999.) i.e. the retroinsular cortex in humans, where extensive visual vestibular convergence has been found. I suggest that the Authors consider the following papers:
Cardin, V. and Smith, A.T., 2010. Sensitivity of human visual and vestibular cortical regions to egomotion-compatible visual stimulation. Cerebral cortex, 20(8), pp.1964-1973.
Frank, S.M. and Greenlee, M.W., 2018. The parieto-insular vestibular cortex in humans: more than a single area? Journal of neurophysiology. 120(3):1438-1450
Indovina, I., Bosco, G., Riccelli, R., Maffei, V., Lacquaniti, F., Passamonti, L. and Toschi, N., 2020. Structural connectome and connectivity lateralization of the multimodal vestibular cortical network. NeuroImage, 222, p.117247.
On the other hand, others have suggested OP2 (part of parietal operculum) as a node of visual vestibular integration in humans:
Ibitoye, R.T., Mallas, E.J., Bourke, N.J., Kaski, D., Bronstein, A.M. and Sharp, D.J., 2023. The human vestibular cortex: functional anatomy of OP2, its connectivity and the effect of vestibular disease. Cerebral Cortex, 33(3), pp.567-582.
Attention has recently been brought to medial regions of the human cortex as implied in visuo-vestibular integration:
Beer, A.L., Becker, M., Frank, S.M. and Greenlee, M.W., 2023. Vestibular and visual brain areas in the medial cortex of the human brain. Journal of Neurophysiology, 129(4), pp.948-962.
Finally, for the brain substrates of visual vestibular integration for navigation in non-primate species, the Authors may want to consider the recent review: Keshavarzi, S., Velez-Fort, M. and Margrie, T.W., 2023. Cortical integration of vestibular and visual cues for navigation, visual processing, and perception. Annual review of neuroscience, 46(1), pp.301-320.
Smith, A.T., Wall, M.B. and Thilo, K.V., 2012. Vestibular inputs to human motion-sensitive visual cortex. Cerebral cortex, 22(5), pp.1068-1077.
Author Response
Dear reviewer,
Thank you very much for your careful and constructive review of our manuscript. We sincerely appreciate your insightful comments and thoughtful suggestions, which have been extremely helpful in improving the clarity, comprehensiveness, and scholarly value of the work.
Please find our point-by-point responses to your comments below. All corresponding changes have been made in the revised manuscript and are marked in blue text and/or tracked for clarity.
Comments and Suggestions for Authors:
This is a very nice, timely review of a complex issue, the neural basis of visual-vestibular integration for self-motion.
In general, the main topics are all well discussed and pertinent literature is cited.
However, there might be some specific integrations that the Authors may want to consider.
I list them following the line numbering of the article.
Authors’ response: We thank the reviewer for the positive comments and thoughtful suggestions, which have greatly helped us refine the manuscript and enhance its scholarly value.
Line 74. “The vestibular system…. struggles to distinguish between head tilt caused by gravity and translation relative to the ground (Angelaki, 2004)”
In fact, work from Angelaki’s group has unraveled some of the basic mechanisms to distinguish head tilt due to gravity and linear translation, and the Authors may want to cite some pertinent work, e.g. Angelaki, D.E., Shaikh, A.G., Green, A.M. and Dickman, J.D., 2004. Neurons compute internal models of the physical laws of motion. Nature, 430(6999), pp.560-564. Laurens, J., Meng, H. and Angelaki, D.E., 2013. Neural representation of orientation relative to gravity in the macaque cerebellum. Neuron, 80(6), pp.1508-1518.
Therefore, it should be clarified that the equivalence problem of gravitational and linear acceleration is addressed by the nervous system not only by integrating visual and vestibular cues, but also by integrating otolith signals and semicircular canals at different stages of the central nervous system (behaviorally, this was already suggested in humans by Merfeld, D.M., Zupan, L. and Peterka, R.J., 1999. Humans use internal models to estimate gravity and linear acceleration. Nature, 398(6728), pp.615-618, as well as by Angelaki, D.E., McHenry, M.Q., Dickman, J.D., Newlands, S.D. and Hess, B.J., 1999. Computation of inertial motion: neural strategies to resolve ambiguous otolith information. Journal of Neuroscience, 19(1), pp.316-327.)
Authors’ response: We thank the reviewer for highlighting these important studies. We have revised the manuscript to clarify that the nervous system resolves the equivalence between gravitational and linear acceleration not only through visual-vestibular integration but also via the central processing of otolith and semicircular canal signals. We have added the suggested key references (Angelaki et al., 2004; Laurens et al., 2013; Merfeld et al., 1999; Angelaki et al., 1999) to provide a more comprehensive overview of the underlying neural mechanisms.
Line 82. This section develops very thoroughly the issue of Bayesian integration. The Authors may want to consider adding this nice review on the specific topic: Angelaki, D.E., Klier, E.M. and Snyder, L.H., 2009. A vestibular sensation: probabilistic approaches to spatial perception. Neuron, 64(4), pp.448-461.
In addition, they may want to consider the issue of visual and vestibular processing of vertical motion by taking into account the effects of gravity: Delle Monache, S., La Scaleia, B., Finazzi Agrò, A., Lacquaniti, F. and Zago, M., 2025. Psychophysical evidence for an internal model of gravity in the visual and vestibular estimates of vertical motion duration. Scientific Reports, 15(1), p.10394.
Authors’ response: We appreciate the reviewer’s insightful suggestions. We have now cited Angelaki et al. (2009) to enrich the discussion on Bayesian integration and probabilistic approaches to spatial perception. Additionally, we have included the recent work by Delle Monache et al. (2025) to highlight how internal models of gravity affect visual and vestibular estimates of vertical motion.
Line 115. “multisensory integration not only depends on the reliability of sensory inputs but is also closely linked to…” It would be better: “not only does multisensory integration depend on…, but it is also..”
Authors’ response: Thank you for the improved phrasing. We have updated that sentence.
Line 227 ref. 47 should be cited since Guldin and Grusser were the first to suggest the term PIVC, at least to the best of my knowledge
Authors’ response: Thank you for pointing this out. We have now cited Guldin and Grüsser’s study at the relevant sentence.
The following lines of the section Brain Regions Involved in Visual-Vestibular Integration give an excellent coverage of the main literature based on work in non-human primates.
However, I suggest that the Authors consider expanding the coverage of literature in humans, since the results may not necessarily align with those in non-human primates.
First, they may want to consider a review comparing monkey and human data. The latter are mainly (but not exclusively, there is also work base on patients) based on neuroimaging for the anatomo-functional substrates in humans. Smith, A.T., Greenlee, M.W., DeAngelis, G.C. and Angelaki, D.E., 2017. Distributed visual–vestibular processing in the cerebral cortex of man and macaque. Multisensory Research, 30(2), pp.91-120.
Authors’ response: We appreciate the reviewer’s valuable suggestion. We have now expanded the coverage of human studies in the revised section and included the recommended review (Smith et al., 2017).
In particular, the question of the human homologue of monkey PIVC has been much debated.
VPS of monkey could be PIC (Sunaert S, Van Hecke P, Marchal G, Orban GA. Motion-responsive regions of the human brain. Exp Brain Res 127: 355–370, 1999.) i.e. the retroinsular cortex in humans, where extensive visual vestibular convergence has been found. I suggest that the Authors consider the following papers:
Cardin, V. and Smith, A.T., 2010. Sensitivity of human visual and vestibular cortical regions to egomotion-compatible visual stimulation. Cerebral cortex, 20(8), pp.1964-1973.
Frank, S.M. and Greenlee, M.W., 2018. The parieto-insular vestibular cortex in humans: more than a single area? Journal of neurophysiology. 120(3):1438-1450
Indovina, I., Bosco, G., Riccelli, R., Maffei, V., Lacquaniti, F., Passamonti, L. and Toschi, N., 2020. Structural connectome and connectivity lateralization of the multimodal vestibular cortical network. NeuroImage, 222, p.117247.
Authors’ response: Thank you for the helpful suggestion. We have added a discussion on the correspondence between monkey VPS and the human PIC region, and have included the recommended references (Sunaert et al., 1999; Cardin and Smith, 2010; Frank and Greenlee, 2018; Indovina et al., 2020) to support this point.
On the other hand, others have suggested OP2 (part of parietal operculum) as a node of visual vestibular integration in humans:
Ibitoye, R.T., Mallas, E.J., Bourke, N.J., Kaski, D., Bronstein, A.M. and Sharp, D.J., 2023. The human vestibular cortex: functional anatomy of OP2, its connectivity and the effect of vestibular disease. Cerebral Cortex, 33(3), pp.567-582.
Authors’ response: Thank you for the suggestion. We have added a discussion on the human OP2 region, highlighting its proposed role in visual–vestibular integration and its correspondence to the macaque PIVC. We have also cited Ibitoye et al. (2023) to elaborate on the functional organization and connectivity of OP2.
Attention has recently been brought to medial regions of the human cortex as implied in visuo-vestibular integration:
Beer, A.L., Becker, M., Frank, S.M. and Greenlee, M.W., 2023. Vestibular and visual brain areas in the medial cortex of the human brain. Journal of Neurophysiology, 129(4), pp.948-962.
Authors’ response: Thank you for highlighting this important work. We have now incorporated recent findings demonstrating the involvement of several medial cortical regions in visual–vestibular processing underlying self-motion perception (Wall and Smith, 2008; Beer et al., 2023). In particular, we discuss the cingulate sulcus visual area (CSv), which shows robust responses to optic flow and vestibular stimulation, supporting its role as a multisensory region.
Finally, for the brain substrates of visual vestibular integration for navigation in non-primate species, the Authors may want to consider the recent review: Keshavarzi, S., Velez-Fort, M. and Margrie, T.W., 2023. Cortical integration of vestibular and visual cues for navigation, visual processing, and perception. Annual review of neuroscience, 46(1), pp.301-320.
Smith, A.T., Wall, M.B. and Thilo, K.V., 2012. Vestibular inputs to human motion-sensitive visual cortex. Cerebral cortex, 22(5), pp.1068-1077.
Authors’ response: Thank you for the suggestion. We have now cited Keshavarzi et al. (2023) and Smith et al. (2012) in the revised manuscript.
Reviewer 2 Report
Comments and Suggestions for Authors
This is a short review of the literature on the topic of visual-vestibular processing of cues related to self-motion perception. The focus is on Bayesian modelling of multisensory information for heading judgments (l. 82 ff). Information is weighted by the reciprocal of the variance of the visual (optic flow) and vestibular (inertial motion of head) signals to derive the most likely estimate of the current heading direction (Fig. 1, p. 5). The main focus is on the neural basis of this multisensory integration in non-human primates (Fig. 2, p.7). Two examples are presented of the directional tuning functions of MSTd and VPS neurons (Fig. 3, p. 10), when the visual and vestibular cues of heading direction are perfectly congruent (0 degrees on the x-axes) or when they become increasing incongruent to this preferred direction ("away" from 0 degrees to the left or to the right). Such multisensory tuning functions point to an enhancement of activation when visual and vestibular direction cues are aligned (ratio: Rcom/Rvis-pref or Rcom/Rves-pref > 1) and this activation is suppressed when the cues are misaligned in direction (ratios < 1). A divisive normalisation model is proposed (Fig. 4, p. 11) to account for these subadditive or suppressive effects on neural activation. As such the review provides a timely summary of the literature in this area. Some recommendations for improvement are listed below.
Major points requiring revision
1) While the review is strongly focused on the neural activity of MSTd and VPS neurons in macaque cortex, there is a vast literature on possible human fMRI correlates of visual-vestibular interactions in simulated self-motion perception. The authors may want to point the reader to reviews of this literature comparing human fMRI with single-unit recordings in macaque brains ( e.g., Lopez & Blanke, 2011; Dieterich & Brandt, 2017; Raiser et al., 2020).
2) While the authors focus on self-motion perception and heading, other important aspects of vestibular inputs that affect visual processing (gaze stabilisation) as well as posture/balance and cognition could be mentioned (with reference to reviews like Hitier et al., 2014; Cullen, 2019).
3) The computational modeling of visual and vestibular cues for heading put forth by other authors should be mentioned (e.g., MacNeilage et al., 2010).
Minor points requiring revision
4) l. 62: Please add citation to Lappe et al., 1999 here.
5) l. 87: Please add citation to MacNeilage et al., 2007 here.
6) l. 227: Please add citation to Grüsser & Guldin (1998) here.
Suggested references
Cullen, K. E. (2019). Vestibular processing during natural self-motion: implications for perception and action. Nature Reviews Neuroscience, 1–18. doi: 10.1038/s41583-019-0153-1
Dieterich, M., & Brandt, T. (2017). Global orientation in space and the lateralization of brain functions. Current Opinion in Neurology, 1–9. doi: 10.1097/wco.0000000000000516
Hitier, M., Besnard, S., & Smith, P. F. (2014). Vestibular pathways involved in cognition. Frontiers in Integrative Neuroscience, 8, 59. doi: 10.3389/fnint.2014.00059
Lappe, M., Bremmer, F., & Berg, A. V. van den. (1999). Perception of self-motion from visual flow. Trends in Cognitive Sciences, 3(9), 329–336. doi: 10.1016/s1364-6613(99)01364-9
Lopez, C., & Blanke, O. (2011). The thalamocortical vestibular system in animals and humans. Brain Research Reviews, 67(1–2), 119–146. doi: 10.1016/j.brainresrev.2010.12.002
MacNeilage, P. R., Banks, M. S., DeAngelis, G. C., & Angelaki, D. E. (2010). Vestibular heading discrimination and sensitivity to linear acceleration in head and world coordinates. The Journal of Neuroscience : The Official Journal of the Society for Neuroscience, 30(27), 9084–9094. doi: 10.1523/jneurosci.1304-10.2010
MacNeilage, P. R., Banks, M. S., Berger, D. R., & Bülthoff, H. H. (2007). A Bayesian model of the disambiguation of gravitoinertial force by visual cues. Experimental Brain Research, 179(2), 263–290. doi: 10.1007/s00221-006-0792-0
Raiser, T. M., Flanagin, V. L., Duering, M., Ombergen, A. van, Ruehl, R. M., & Eulenburg, P. zu. (2020). The human corticocortical vestibular network. NeuroImage, 223, 117362. doi: 10.1016/j.neuroimage.2020.117362
Author Response
Dear reviewer,
We sincerely thank you for your careful and insightful review of our manuscript. Your constructive suggestions and thoughtful comments have greatly contributed to improving the quality of our work. We truly appreciate the time and effort you have devoted to this review.
Please find below our detailed responses to each of your comments. Corresponding revisions have been made in the manuscript, with all changes clearly indicated in blue text and/or tracked for your convenience.
Comments and Suggestions for Authors:
This is a short review of the literature on the topic of visual-vestibular processing of cues related to self-motion perception. The focus is on Bayesian modelling of multisensory information for heading judgments (l. 82 ff). Information is weighted by the reciprocal of the variance of the visual (optic flow) and vestibular (inertial motion of head) signals to derive the most likely estimate of the current heading direction (Fig. 1, p. 5). The main focus is on the neural basis of this multisensory integration in non-human primates (Fig. 2, p.7). Two examples are presented of the directional tuning functions of MSTd and VPS neurons (Fig. 3, p. 10), when the visual and vestibular cues of heading direction are perfectly congruent (0 degrees on the x-axes) or when they become increasing incongruent to this preferred direction ("away" from 0 degrees to the left or to the right). Such multisensory tuning functions point to an enhancement of activation when visual and vestibular direction cues are aligned (ratio: Rcom/Rvis-pref or Rcom/Rves-pref > 1) and this activation is suppressed when the cues are misaligned in direction (ratios < 1). A divisive normalisation model is proposed (Fig. 4, p. 11) to account for these subadditive or suppressive effects on neural activation. As such the review provides a timely summary of the literature in this area. Some recommendations for improvement are listed below.
Major points requiring revision
1) While the review is strongly focused on the neural activity of MSTd and VPS neurons in macaque cortex, there is a vast literature on possible human fMRI correlates of visual-vestibular interactions in simulated self-motion perception. The authors may want to point the reader to reviews of this literature comparing human fMRI with single-unit recordings in macaque brains ( e.g., Lopez & Blanke, 2011; Dieterich & Brandt, 2017; Raiser et al., 2020).
Authors’ response: Thank you for the helpful suggestion. We have expanded the discussion in Section 5.1 to include comparisons between human fMRI studies and single-unit recordings in macaques, with a particular focus on identifying homologous brain regions involved in visual–vestibular integration. In addition, we have cited relevant review articles (Lopez & Blanke, 2011; Dieterich & Brandt, 2017; Raiser et al., 2020) to guide readers toward comprehensive summaries of the human neuroimaging literature and its correspondence with findings from non-human primate research.
2) While the authors focus on self-motion perception and heading, other important aspects of vestibular inputs that affect visual processing (gaze stabilisation) as well as posture/balance and cognition could be mentioned (with reference to reviews like Hitier et al., 2014; Cullen, 2019).
Authors’ response: Thank you for the insightful comment. In response, we have added a paragraph highlighting the broader functional roles of vestibular inputs beyond self-motion perception and heading. Specifically, we now discuss their critical contributions to gaze stabilization, postural control, and higher-level cognitive functions such as spatial orientation and navigation. We have also cited the suggested reviews (Hitier et al., 2014; Cullen, 2019) to provide readers with a comprehensive overview of these additional aspects of vestibular processing.
3) The computational modeling of visual and vestibular cues for heading put forth by other authors should be mentioned (e.g., MacNeilage et al., 2010).
Authors’ response: Thank you for the suggestion. We have now cited the computational model proposed by MacNeilage et al. (2010).
Minor points requiring revision
4) l. 62: Please add citation to Lappe et al., 1999 here.
Authors’ response: Thank you for the suggestion. We have now added the citation to Lappe et al. (1999) at the indicated location.
5) l. 87: Please add citation to MacNeilage et al., 2007 here.
Authors’ response: Thank you for pointing this out. We have added the citation to MacNeilage et al. (2007) at the indicated location.
6) l. 227: Please add citation to Grüsser & Guldin (1998) here.
Authors’ response: Thank you for the reminder. We have now cited Grüsser and Guldin (1998) to acknowledge their foundational work in defining PIVC.
Suggested references
Cullen, K. E. (2019). Vestibular processing during natural self-motion: implications for perception and action. Nature Reviews Neuroscience, 1–18. doi: 10.1038/s41583-019-0153-1
Dieterich, M., & Brandt, T. (2017). Global orientation in space and the lateralization of brain functions. Current Opinion in Neurology, 1–9. doi: 10.1097/wco.0000000000000516
Hitier, M., Besnard, S., & Smith, P. F. (2014). Vestibular pathways involved in cognition. Frontiers in Integrative Neuroscience, 8, 59. doi: 10.3389/fnint.2014.00059
Lappe, M., Bremmer, F., & Berg, A. V. van den. (1999). Perception of self-motion from visual flow. Trends in Cognitive Sciences, 3(9), 329–336. doi: 10.1016/s1364-6613(99)01364-9
Lopez, C., & Blanke, O. (2011). The thalamocortical vestibular system in animals and humans. Brain Research Reviews, 67(1–2), 119–146. doi: 10.1016/j.brainresrev.2010.12.002
MacNeilage, P. R., Banks, M. S., DeAngelis, G. C., & Angelaki, D. E. (2010). Vestibular heading discrimination and sensitivity to linear acceleration in head and world coordinates. The Journal of Neuroscience : The Official Journal of the Society for Neuroscience, 30(27), 9084–9094. doi: 10.1523/jneurosci.1304-10.2010
MacNeilage, P. R., Banks, M. S., Berger, D. R., & Bülthoff, H. H. (2007). A Bayesian model of the disambiguation of gravitoinertial force by visual cues. Experimental Brain Research, 179(2), 263–290. doi: 10.1007/s00221-006-0792-0
Raiser, T. M., Flanagin, V. L., Duering, M., Ombergen, A. van, Ruehl, R. M., & Eulenburg, P. zu. (2020). The human corticocortical vestibular network. NeuroImage, 223, 117362. doi: 10.1016/j.neuroimage.2020.117362
Authors’ response: Thank you for providing the list of references. We have now incorporated all the recommended citations into the revised manuscript where appropriate.
Round 2
Reviewer 1 Report
Comments and Suggestions for Authors
The Authors have thoroughly revised their ms which now is fully satistfactory
Reviewer 2 Report
Comments and Suggestions for Authors
The authors have followed my suggestions for revision and they have made appropriate changes to the revised manuscript. The inclusion of the results of brain-imaging studies in humans adds to the scope of the review.